# Enhancing Fine-grained Multi-modal Alignment via Adapters: A Parameter-Efficient Training Framework for Referring Image Segmentation

Zunnan Xu [* 1]   Jiaqi Huang [* 1]   Ting Liu [† 2]   Yong Liu [1]   Kehong Yuan [‡ 1]   Xiu Li [‡ 1]

## Abstract

In the domain of computer vision, Parameter-Efficient Training (PET) is increasingly replacing the traditional paradigm of pre-training followed by full fine-tuning. PET is particularly favored for its effectiveness in large scale models, as it streamlines transfer learning costs and optimizes hardware utilization. However, the prevailing PET methods are primarily designed for single-modal optimization without fine-grained feature extraction design. When applied to multi-modal dense prediction tasks, these methods typically do not match the performance of full fine-tuning methods that utilize more resources. In this paper, we do an investigation of efficient training problems on referring image segmentation. We introduce DenseCrossAdapter, a parameter-efficient module designed to enhance low-rank visual feature propagation by establishing dense interconnections between each layer and all preceding layers. This facilitates robust cross-modal feature interaction. We also suggest using text adapters to improve textual features. Our approach greatly surpasses state-of-the-art methods with only 0.9% to 1.8% backbone parameter updates, evaluated on challenging benchmarks.

## 1. Introduction

In the realm of computer vision, Parameter-Efficient Training (PET) is rapidly emerging as a superior alternative to the traditional approach of pre-training followed by full fine-tuning. PET is particularly advantageous for large-scale models because it significantly reduces the costs associated with transfer learning and optimizes the utilization of computational resources.

Despite their benefits, current PET methods are often limited to single-modal optimization and lack a fine-grained feature extraction design. This limitation becomes apparent when these methods are applied to multi-modal dense prediction tasks. They typically fail to match the performance of full fine-tuning methods that leverage more extensive resources (Liu et al., 2024c). One such challenging task is Referring Image Segmentation (RIS), which involves predicting the mask of a target object within an image based on a natural language description. Unlike semantic segmentation, which assigns predefined labels to each pixel, RIS requires a more sophisticated understanding of both language and visual content to accurately identify the described object. The significance of RIS lies in its ability to bridge the gap between natural language descriptions and fine-grained visual perception (Liu et al., 2024b; Ji et al., 2024). This capability is pivotal for the advancement of artificial intelligence, especially in autonomous systems, image-based retrieval, and human-computer interaction. The complexity of RIS stems from the need to interpret variable context lengths and to understand an open-world vocabulary encompassing a diverse range of object names, attributes, and positional references (Li et al., 2024b; Wu* et al., 2024). The requirement for precise segmentation of referring objects elevates this dense prediction task to one of the most formidable challenges of vision-language understanding.

For these challenging vision-language tasks, a prevalent trend is to scale up foundational models (Radford et al., 2021; Li et al., 2022b; Oquab et al., 2023; Fang et al., 2023). These models leverage large datasets to learn a comprehensive set of visual features. In return, the scale-up process not only enhances their ability to discern subtleties in visual data but also significantly boosts their generalization capabilities, demonstrating a robustness that is essential for real-world applications. However, there often exists a gap between the pre-trained tasks of these models and the specific requirements of downstream applications. Bridging this gap through efficient adaptation presents a formidable challenge. Recent studies (Wang et al., 2022; Ding et al., 2022; Yang et al., 2022; Liu et al., 2023a) have demonstrated

---

[*]Equal contribution. [†]Work done during Ting Liu's visit at Tsinghua University. [‡]Corresponding authors. [1]SIGS, Tsinghua University [2]Tsinghua University. Correspondence to: Xiu Li <li.xiu@sz.tsinghua.edu.cn>, Kehong Yuan <yuankh@sz.tsinghua.edu.cn>.

Accepted to the Workshop on Advancing Neural Network Training at International Conference on Machine Learning (WANT@ICML 2024).

the effectiveness of fine-tuning powerful pre-trained models for referring image segmentation. However, a common challenge is that they typically require full fine-tuning to adapt to dense prediction tasks. This process can lead to the loss of valuable pre-training knowledge, as it involves adjusting a large number of parameters that were previously optimized during the pre-training phase (Kim et al., 2022; Liu et al., 2023a;b). Moreover, these approaches maintain a distinct set of fine-tuned parameters of pre-trained models for each dataset, which can lead to substantial deployment costs. The problem becomes particularly serious when considering the ever-growing size of pre-trained models, which now include parameters ranging from hundreds of millions to trillions (Li et al., 2022a; Zhou et al., 2022; Chen et al., 2022b; Sun et al., 2023).

Considering the above problems, we extend the discussion of ETRIS (Xu et al., 2023): *Is it possible for a model, with its pre-trained backbone parameters fixed, to surpass the performance of existing full fine-tuned methods?* Various parameter-efficient training methods have been developed to achieve an optimal equilibrium between operational efficiency and model performance (Gao et al., 2021; Chen et al., 2022a; Zhou et al., 2022; Wang et al., 2023; Li et al., 2023; Liu et al., 2024b). However, despite these contributions, most existing methods are limited in their scope, predominantly applied to single-modality tasks (Guo et al., 2020; Houlsby et al., 2019; Chen et al., 2022c) or simple classification problems (Gao et al., 2021; Chen et al., 2022a; Zhou et al., 2022). There remains a notable gap concerning dense prediction tasks and the nuanced interactions between multiple modalities. Pioneering works like ETRIS and BarleRIa (Wang et al., 2023) aimed to parameter-efficient fine-tuning CLIP (Radford et al., 2021) on referring image segmentation, but they faced several limitations: (i) These methods primarily relied on the early-stage fusion of multimodal features from the backbone, missing out on the benefits of more comprehensive global features, leading to suboptimal results. (ii) Furthermore, existing parameter-efficient modules, such as Bridger (Xu et al., 2023) and GST (Wang et al., 2023), are constrained by their limited application of multi-scale modeling. Their approach, which focuses on adjusting channel dimensions and using multi-head attention mechanisms, is insufficient for capturing the full complexity of visual data across different scales.

Our method addresses this question by introducing a novel approach that enhances the effectiveness of adapting pre-trained vision-language models. In detail, we propose an adapter named DenseCrossAdapter, which can be seamlessly integrated into the pre-trained model for dense prediction tasks. There are two tailored modules for DenseCrossAdapter: (i) a densely connected prior module for capturing the local multi-scale semantics feature maps of the intermediate layer and (ii) a cross-modal attention module that enables information exchange between visual and textual features. Secondly, we propose incorporating text adapters to enhance the text encoder. We further leverage these enhanced features to improve alignment between visual and linguistic features.

Our framework is constructed around a dual-encoder architecture. Unlike previous methods, we have selected DINO (Oquab et al., 2023) to act as our visual encoder. The reason we chose DINO as our vision backbone is based on several insights: (i) DINO's self-supervised learning approach provides robust generalization and is more advantageous for dense prediction tasks compared to CLIP (Radford et al., 2021). (ii) The lack of multimodal pre-training in DINO, notably in visual-text alignment, poses challenges for its direct application on referring image segmentation. This gap highlights the essential role of our proposed module in enhancing the model's capabilities, particularly in improving fine-grained vision language alignment. Our main contributions are as follows:

- We introduce the pre-trained model DINO in RIS tasks and provide an effective training strategy for fine-grained alignment that avoids the need for intricate design.
- We propose a novel DenseCrossAdapter that can be seamlessly integrated into the pre-trained backbone to enhance and interact with its intermediate features. This integration enhances DINO's alignment with language and improves its performance on dense prediction tasks.
- Experiments demonstrate that our method greatly surpasses state-of-the-art full fine-tuned methods in referring image segmentation, with only 0.9% to 1.8% backbone parameter updates.

## 2. Related Work

**Parameter Efficient Training (PET)** aims to streamline the process of adapting pre-trained models to new tasks with minimal parameter adjustments, making it a practical solution for deploying large models to individual users, particularly in the face of expanding model sizes. Previous PET methods can be mainly divided into three types: (i) updating newly added parameters to the model or input (Houlsby et al., 2019; Li & Liang, 2021; Zhou et al., 2022; Li et al., 2024a; Liu et al., 2024a); (ii) sparsely updating a small number of parameters of the model (Guo et al., 2020; Zaken et al., 2021); (iii) low-rank factorization for the weights to be updated (Hu et al., 2021; Karimi Mahabadi et al., 2021; Hao et al., 2023). However, previous works applying PET in computer vision mainly focus on classification and generation tasks. How to efficiently update and transfer the pre-trained knowledge space to dense prediction tasks remains a great challenge. Pioneering works like ETRIS (Xu et al., 2023) and BarleRIa (Wang et al., 2023) sought to utilize adapters to fine-tune CLIP (Radford et al., 2021)

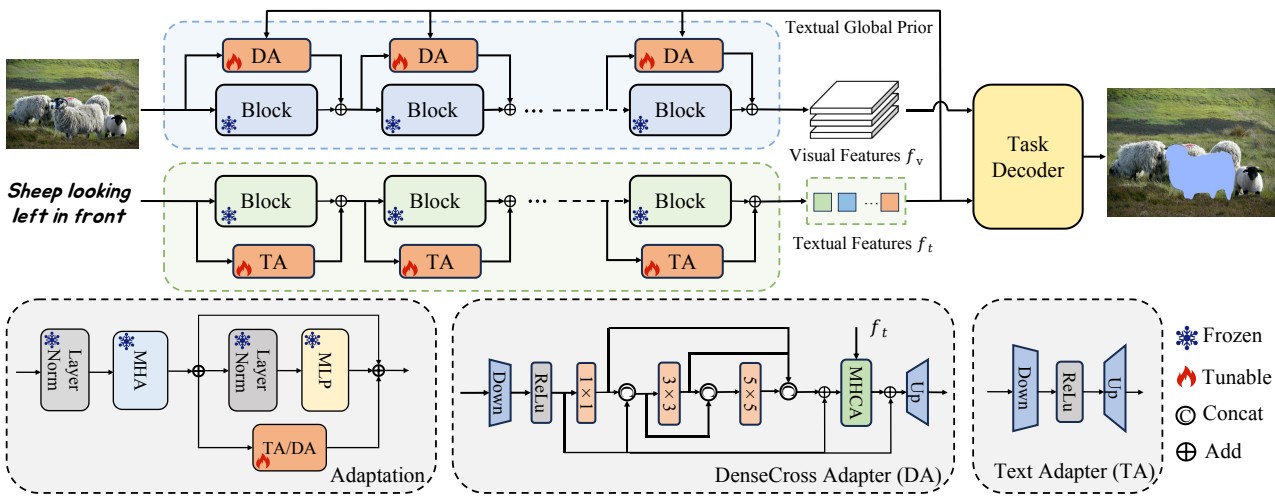

*Figure 1.* Overall framework of our DCRIS. In the text branch, we obtain the text feature $f_t$ using text blocks with Text Adapters (TA). In the image branch, we utilize DenseCross Adapters (DA) to facilitate cross-modal and multi-scale modeling of low-rank visual features. This approach incorporates textual global prior information to enhance the visual features $f_v$.

for referring image segmentation. However, their proposed modules like Bridger (Xu et al., 2023) and GST (Wang et al., 2023) are insufficient for capturing the complexity of multi-scale visual features.

**Referring Image Segmentation (RIS)** aims to segment the target objects referred to by natural language descriptions. It necessitates the models to comprehensively associate diverse visual content and linguistic signals. The genesis of this field can be traced to CNN-LSTM-based methods, such as the Referring Relation Network (RRN) (Li et al., 2018) and the Recurrent Multimodal Interaction (RMI) (Liu et al., 2017). These early methods used CNN and LSTM networks to separately extract visual and linguistic features, which were then combined to form cross-modal representations for segmentation by an FCN. The advent of the Transformer model has catalyzed a paradigm transformation of integrating features across diverse modalities by attention mechanism (Yang et al., 2022; Liu et al., 2023a; Yan et al., 2023; Liu et al., 2023b). Among them, MDETR (Kamath et al., 2021) and VLT (Ding et al., 2022) have demonstrated remarkable performance across various Vision-Language (VL) tasks by integrating multi-modal attention interaction and query representation. Capitalizing on the robust image-text alignment capabilities of CLIP, CRIS (Wang et al., 2022), ETRIS (Xu et al., 2023) and UniLSeg (Liu et al., 2023c) zeroes in on sentence-pixel alignment to harness the wealth of multi-modal correspondences. However, existing methods primarily concentrate on the design of visual-linguistic interactions during the decoding phase, while overlooking the potential to fully excavate the pretrained backbone networks. To this end, we propose DCRIS, a framework that aligns features from different modalities

with the assistance of parameter-efficient modules boosting multi-scale comprehensive updating of backbone networks. Compared to existing fully fine-tuning methods, the proposed approach achieves competitive performance while greatly reducing the training overhead.

## 3. Methodology

### 3.1. Framework Overview

The overall framework of the proposed model is illustrated in Figure 1. Our approach freezes the parameters of the pretrained backbone, ensuring parameter efficiency. The core of the model's design philosophy is the DenseCrossAdapter, which is intended to facilitate interaction between cross-modal features and inject dense prediction priors into the pre-trained backbone. This is coupled with the incorporation of text adapters in the text encoder to enhance fine-grained image text alignments.

### 3.2. Image & Text Feature Extraction

**Visual encoder.** Our work adapts the distilled DINOv2 with registers (Oquab et al., 2023) as the backbone. This model has been well-pretrained on self-supervised training tasks and is based on ViT-B/14. Specifically, for an input image $I \in R^{H \times W \times 3}$, we utilize DINOv2 enhanced with densecross adapters to extract image features. We select the outputs of several middle layers and the last layer as hierarchical vision features $f_v^i, i \in [1, 2, 3]$, which are used for subsequent dense prediction modules.

**Text encoder.** For the input refer expression $T$, we utilize

the pre-trained text transformer of CLIP (Radford et al., 2021) for extracting text features. Text features $f_t$ and sentence-level feature $f_s$ are extracted using a CLIP text encoder, augmented with text adapters to guide referring image segmentation. Considering the substantial number of parameters the encoder occupies, and to avoid the loss of valuable pre-training knowledge, we freeze all the encoder parameters during our fine-tuning method to apply it efficiently in downstream tasks.

### 3.3. Local & Global Feature Interaction

As mentioned in Section 1, although DINOv2 has strong generalization capabilities and is more advantageous than CLIP in tasks that rely more on visual abilities, DINOv2 lacks visual-text alignment in the RIS downstream task due to the absence of multimodal pre-training. Meanwhile, the self-supervising tasks of DINOv2 are limited to patch-level, lacking pixel-level information in dense prediction tasks. To address this and enhance the model's multi-scale modeling capability, we designed and utilized densecross adapters to augment the model's vision backbone. The visual backbone network remains fixed, with training solely focused on the DenseCross adapter parameters.

**DenseCrossAdapter.** As shown in Figure 1, the proposed DenseCross Adapter mainly differs from previous adapter designs by integrating a densely connected prior module. A cross-modal attention module is included between the activation and up-projection layers. This integration enhances the model's capability to extract dense image features and enriches its multimodal perception abilities. In the densely connected module, a multi-branch convolutional structure is proposed to achieve multi-scale modeling of low-rank visual features and acquire multi-scale prior information. After applying linear projection and non-linear activation, $1 \times 1$, $3 \times 3$, and $5 \times 5$ convolutional kernels are used to progressively integrate outputs from the previous layer. To ensure the module's lightweight and efficiency, $1 \times 1$ convolutions are also used before the $3 \times 3$ and $5 \times 5$ convolutions to reduce channel dimensions. The output of the $1 \times 1$ convolution serves as the input for the $3 \times 3$ convolution, and the combined outputs of the $1 \times 1$ and $3 \times 3$ convolutions are used as the input for the $5 \times 5$ convolution. Unlike the typical approach of directly merging multi-scale features, this stepwise integration method intricately combines fine details with broader contextual information. To maintain the integrity of the original features, the initial features are added to the final concatenated features.

Denote $\sigma$ as a non-linear activation function (ReLU), $\text{Linear}_{\text{down}}$ as a downsampling operation of linear projection. Specifically, for given input image features $f_v^l$ at layer $l$, this

process can be formulated as below:

$$
\begin{aligned}
F_v^l &= \sigma(\text{Linear}_{\text{down}}(f_v^l)), \\
F_{v1}^l &= \text{conv}_{1 \times 1}(F_v^l), \\
F_{v2}^l &= \text{conv}_{3 \times 3}(F_v^l, F_{v1}^l), \\
F_{v3}^l &= \text{conv}_{5 \times 5}(F_v^l, F_{v1}^l, F_{v2}^l), \\
F_{\text{dense}}^l &= (F_{v1}^l, F_{v2}^l, F_{v3}^l) + F_v^l,
\end{aligned}
\tag{1}
$$

where $F_{v1}^l$, $F_{v2}^l$, and $F_{v3}^l$ are the output features after applying respective convolutional operations, and $F_{\text{dense}}^l$ is the final dense feature representation obtained by concatenating the features from all branches and adding the initial feature $F_v^l$. Here, $(,)$ represents concatenating the features along the dimensional direction.

Considering that textual information contains valuable references, we utilize it as a global reference prior by integrating it into the vision backbone network via a multi-head cross-attention mechanism. This not only regularizes the visual features but also aligns them better with the extracted text features (denoted as $f_t$). This process can be formalized as:

$$
\begin{aligned}
F_{\text{cross}}^l &= \mathcal{F}_{\text{MHCA}}(F_{\text{dense}}^l, f_t) + F_v^l, \\
f_{dc}^l &= \text{Linear}_{\text{up}}(F_{\text{cross}}^l),
\end{aligned}
\tag{2}
$$

where $\mathcal{F}_{\text{MHCA}}$ denotes multi-head cross attention, $F_{\text{cross}}^l$ denotes the fused visual feature, and $\text{Linear}_{\text{up}}$ represents an operation to project the visual features back to get $f_{dc}^l$. We add the DenseCrossAdapter in parallel to the MLP layer in the transformer block as illustrated in Figure1.

**Text adapter.** Acknowledging the disparities between the text features extracted by the CLIP text encoder and the DINO features, we incorporate a text adapter to improve the text encoder for fine-grained alignment of linguistic and visual features. In contrast to the DenseCross adapter, the text adapter adopts a more straightforward design, focusing on effectively processing and integrating text features without complex structures. This minimalist approach ensures efficient handling of textual information while maintaining compatibility with the overall model architecture. By utilizing these enhanced features, we can further improve the alignment between visual and language features. Specifically, for given input text features $f_t^l$ at layer $l$, the text adapter employs a standard "Down-ReLU-Up" structure to refine and project linguistic features. This process can be formalized as:

$$
\begin{aligned}
F_t^l &= \text{Linear}_{\text{down}}(f_t^l), \\
F_{\text{relu}}^l &= \text{ReLU}(F_t^l), \\
f_{\text{w}}^l &= \text{Linear}_{\text{up}}(F_{\text{relu}}^l),
\end{aligned}
\tag{3}
$$

where $\text{Linear}_{\text{down}}$ represents a downsampling linear projection and $\text{Linear}_{\text{up}}$ denotes an upsampling operation to adapt text features back to the original dimension.

## 3.4. The Referring Image Segmentation Head

Following CRIS(Wang et al., 2022) and ETRIS(Xu et al., 2023), we incorporate a learnable referring image segmentation head, which consists of three main components: a cross-modal neck, a vision-language decoder, and an up-sample projector. These components work together to extract the cross-modal feature $F_c$ and the textual feature $F_l$.

The cross-modal neck takes multiple adapted visual features ($\hat{f}_v^i, i \in [1,2,3]$) from three layers of the visual encoder (e.g., the 1/3, 2/3, and the last layer of the backbone) and the adapted textual embeddings $\hat{f}_t$. Specifically, we employ a multi-head cross-attention mechanism ($\mathcal{F}_{\text{MHCA}}$) with convolution to fuse these features, obtaining the fusion features $F_f$. Subsequently, we concatenate a 2D spatial coordinate feature $F_{\text{coord}}$ with $F_f$ and further fuse them using a 3×3 convolution, which can be formalized as:

$$f_c = \text{Conv}([F_f, F_{\text{coord}}]), \qquad (4)$$

where $F_f = \mathcal{F}_{\text{MHCA}}(\hat{f}_v^i, \hat{f}_t)$ and $f_c$ denotes the combined cross-modal feature.

The vision-language decoder further merges the composite feature $f_c$ with the textual embeddings $\hat{f}_t$. This fusion process culminates in the generation of multimodal features $F_{mm}$, encapsulating both visual and linguistic information. Specifically, the decoder consists of three layers, each composed of a multi-head self-attention layer (MHSA), a multi-head cross-attention layer (MHCA), and a feed-forward network. Within each decoder layer, the combined features $f_c$ are fed into the MHSA layer to capture global contextual information. The MHCA layer further facilitates multi-modal interaction by mapping visual features to queries and textual features to keys and values. Following the MHCA layer, an MLP block, along with layer normalization and residual connections, further processes the output features.

An up-sampling projector further transforms the multimodal features $F_{mm}$ and the sentence-level feature $f_s$ to extract the cross-modal feature $F_c$ and the transformed textual feature $F_l$. $f_s$ is first transformed into $F_l$ through a linear transformation, then split and reshaped into weights and bias, enabling it to function as a Conv2D layer. This Conv2D layer is used to transform the cross-modal representation into the final mask prediction. The overall transformation is achieved using a 4× upsampling followed by convolution and linear projection:

$$\begin{aligned} F_c &= \text{Conv}(\text{UpSample}(F_{mm})), \\ F_l &= \text{Linear}(f_s). \end{aligned} \qquad (5)$$

## 3.5. Training Objective

For the training objective of our model, we utilize a text-to-visual contrastive loss, denoted as $\mathcal{L}_{\text{con}}$, to optimize the alignment between the text-derived features and their corresponding visual pixels. This contrastive loss is designed to both enhance the connection between text features and corresponding visual pixels, and separate these text features from any unrelated visual elements. The text-to-pixel contrastive loss is calculated as below:

$$\begin{aligned} \mathcal{L}_{\text{con}}^i\left(F_c^i, F_l\right) &= \begin{cases} -\log\left(\sigma\left(F_c^i \cdot F_l\right)\right), & i \in \mathcal{P} \\ -\log\left(1 - \sigma\left(F_c^i \cdot F_l\right)\right), & i \in \mathcal{N} \end{cases} \\ \mathcal{L}_{\text{con}}\left(F_c, F_l\right) &= \frac{1}{|\mathcal{P} \cup \mathcal{N}|} \sum_{i \in \mathcal{P} \cup \mathcal{N}} \mathcal{L}_{\text{con}}^i\left(F_c^i, F_l\right), \end{aligned} \qquad (6)$$

where $\mathcal{P}$ and $\mathcal{N}$ denote the class of 1 and 0 in the ground truth, and $\sigma$ denotes the sigmoid function. The loss thus penalizes incorrect alignments between features and encourages the model to correctly match textual descriptions to their associated visual representations.

# 4. Experiments

## 4.1. Datasets

We employ three challenging referring image segmentation benchmarks in our experiments:

- **RefCOCO (Kazemzadeh et al., 2014)** is widely used as a benchmark for referring image segmentation. It comprises 19,994 images annotated with 142,210 referring expressions for 50,000 objects, which have been sourced from the MSCOCO dataset through a two-player game. The dataset is divided into four subsets, consisting of 120,624 training samples, 10,834 validation samples, 5,657 samples for testA, and 5,095 samples for testB, respectively. The average length of the expressions is 3.6 words, and each image contains a minimum of two objects.
- **RefCOCO+ (Kazemzadeh et al., 2014)** dataset consists of 141,564 expressions with 49,856 objects in 19,992 images, which is divided into four subsets: 120,624 train, 10,758 validation, 5,726 testA, and 4,889 testB. Notably, the RefCOCO+ dataset has been constructed to be more challenging than the RefCOCO dataset by excluding certain types of absolute location words.
- **G-Ref (Yu et al., 2016)** comprises 104,560 referring expressions associated with 54,822 objects in 26,711 images. The expressions in G-Ref were collected from Amazon Mechanical Turk and had an average length of 8.4 words, which included more words related to locations and appearances. We present results for both the Google and UMD partitioning methods for G-Ref.

## 4.2. Implementation Details

In our experiments, we use the DINOv2-B/14 as the vision backbone for DCRIS-B, and DINOv2-L/14 as the vision backbone for DCRIS-L. Both models employ the text en-

*Table 1.* State-of-the-art comparison of RIS methods and the PET RIS method on RefCOCO/RefCOCO+/G-Ref datasets without using extra data and Mixed RefCOCO dataset, evaluated using the IoU metric. For Mixed RefCOCO datasets, models marked with * are tuned using the mixed RefCOCO/RefCOCO+/G-Ref datasets. The best results are in bold.

| Method | RefCOCO | | | RefCOCO+ | | | G-Ref | | | Avg |
|---|---|---|---|---|---|---|---|---|---|---|
| | val | testA | testB | val | testA | testB | val(u) | test(u) | val(g) | |
| Traditional Full Fine-tuning | | | | | | | | | | |
| RRN[CVPR 18] (Li et al., 2018) | 55.3 | 57.3 | 54.0 | 39.8 | 42.2 | 36.1 | - | - | 36.5 | 43.8 |
| MAttNet[CVPR 18] (Yu et al., 2018) | 56.5 | 62.4 | 51.7 | 46.7 | 52.4 | 40.1 | 47.6 | 48.6 | - | 50.5 |
| CMSA[CVPR 19] (Ye et al., 2019) | 58.3 | 60.6 | 55.1 | 43.8 | 47.6 | 37.9 | - | - | 40.0 | 47.0 |
| CAC[BMVC 19] (Chen et al., 2019) | 58.9 | 61.8 | 53.8 | - | - | - | 46.4 | 47.0 | 44.3 | - |
| BRINet[CVPR 20] (Hu et al., 2020) | 61.4 | 63.4 | 59.6 | 48.6 | 52.9 | 42.1 | - | - | 48.0 | 52.5 |
| CMPC+[TPAMI 21] (Liu et al., 2021) | 61.4 | 64.5 | 59.6 | 49.6 | 53.4 | 43.2 | - | - | - | - |
| CGAN[ACMMM 20] (Luo et al., 2020) | 64.9 | 68.0 | 62.1 | 51.0 | 55.5 | 44.1 | 51.0 | 51.7 | - | 55.5 |
| LTS[CVPR 21] (Jing et al., 2021) | 65.4 | 67.8 | 63.1 | 54.2 | 58.3 | 48.0 | - | - | - | - |
| VLT[ICCV 21] (Ding et al., 2022) | 65.7 | 68.3 | 62.7 | 55.5 | 59.2 | 49.4 | - | - | 49.8 | 56.7 |
| ReSTR[CVPR 22] (Kim et al., 2022) | 67.2 | 69.3 | 64.5 | 55.8 | 60.4 | 48.3 | 54.5 | - | 54.5 | 58.8 |
| CRIS[CVPR 22] (Wang et al., 2022) | 70.5 | 73.2 | 66.1 | 62.3 | 68.1 | 53.7 | 59.9 | 60.4 | - | 63.8 |
| LAVT[CVPR 22] (Yang et al., 2022) | 72.7 | 75.8 | 68.8 | 62.1 | 68.4 | 55.1 | - | - | 60.5 | 64.9 |
| SEEM[NeurIPS 23] (Zou et al., 2023) | - | - | - | - | - | - | 65.6 | - | - | - |
| VPD[ICCV 23] (Zhao et al., 2023) | 73.5 | - | - | 63.9 | - | - | 63.1 | - | - | 66.8 |
| ReLA[CVPR 23] (Liu et al., 2023a) | 73.8 | 76.5 | 70.18 | 66.0 | 71.0 | 57.7 | 65.0 | 66.0 | 62.7 | 67.5 |
| CGFormer[CVPR 23] (Tang et al., 2023) | 74.8 | 77.3 | 70.6 | 64.5 | 71.0 | 57.1 | 64.7 | 65.1 | 62.5 | 67.7 |
| LISA-7B[CVPR 24] (Lai et al., 2023) | 74.1 | 76.5 | 71.1 | 62.4 | 67.4 | 56.5 | 66.4 | 68.5 | - | 67.9 |
| MagNet[CVPR 24] (Chng et al., 2023) | 75.2 | 78.2 | 71.1 | 66.2 | 71.3 | 58.1 | 65.4 | 66.2 | 63.1 | 68.3 |
| Parameter Efficient-Training | | | | | | | | | | |
| ETRIS[ICCV 23] (Xu et al., 2023) | 70.5 | 73.5 | 66.6 | 60.1 | 66.9 | 50.2 | 59.8 | 59.9 | 57.9 | 62.8 |
| BarLeRIa[ICLR 24] (Wang et al., 2023) | 72.4 | 75.9 | 68.3 | 65.0 | 70.8 | 56.9 | 63.4 | 63.8 | 61.6 | 66.5 |
| DCRIS-B (Ours) | 76.4 | 77.9 | 73.7 | 69.8 | 74.4 | 62.6 | 68.4 | 68.5 | 65.9 | 70.8 |
| DCRIS-L (Ours) | **77.9** | **79.0** | **75.0** | **71.9** | **75.9** | **65.7** | **70.1** | **70.9** | **68.5** | **72.8** |
| With Mixed Training Data | | | | | | | | | | |
| PolyFormer-L*[CVPR 23] (Liu et al., 2023b) | 76.0 | 78.3 | 73.3 | 69.3 | 74.6 | 61.9 | 69.2 | 70.2 | - | 71.6 |
| UNINEXT-L*[CVPR 23] (Yan et al., 2023) | 80.3 | 82.6 | 77.8 | 70.0 | 74.9 | 62.6 | 73.4 | 73.7 | - | 74.4 |
| DCRIS-L* (Ours) | **80.7** | **82.9** | **77.9** | **73.3** | **78.3** | **65.8** | **74.1** | **75.5** | - | **76.1** |

coder from CLIP (Radford et al., 2021) as the textual encoder, with input images resized to 448x448 pixels. The DenseCross Adapter with a dimension of 128 is applied at layers [1, 3, 5, 7, 9, 11] in DCRIS-B and at layers [2, 6, 10, 14, 18, 22] in DCRIS-L. The text adapter with a dimension of 64 is applied at layers [1, 3, 5, 7, 9, 11] in both models. We train the entire framework for 50 epochs using the Adam optimizer with an initial learning rate of 0.0001. The learning rate decays by a factor of 0.1 at epochs 35. Specifically, DCRIS-B is trained on 2 A100 GPUs with a batch size of 32, while DCRIS-L is trained on 4 A100 GPUs with a batch size of 64 and an initial learning rate of 0.0002. Following previous methods (Yang et al., 2022; Tang et al., 2023), we adopt IoU as the metric to evaluate performance, which calculates the intersection regions over the union regions of the predicted segmentation mask and the ground truth.

### 4.3. Main Results

We conducted a comprehensive comparison between our proposed DCRIS models and a series of previous referring image segmentation (RIS) methods. As demonstrated in Table 1, our approach significantly outperforms state-of-the-art RIS methods on three commonly challenging benchmarks. DCRIS-B achieves an average IoU score of 70.8, while DCRIS-L achieves an average IoU score of 72.8, representing improvements of 3.7% and 6.6% compared to the previous state-of-the-art method. Notably, our DCRIS-L model achieves the highest scores across all evaluation tasks, with strong performance on the RefCOCO+ and G-Ref datasets, which present greater challenges compared to RefCOCO.

In addition to comparing against full fine-tuning methods, we also evaluated our models in the context of parameter-efficient training methods. Table 1 shows that DCRIS-B

*Table 2.* Comparison of Parameter-Efficient Tuning Methods Using DINO-B as Backbone on RefCOCO. To ensure fairness, we kept the original parameter settings from prior methods and also adjusted the size of rank to achieve comparable parameter counts.

| Method | RefCOCO | | | Avg | Parameters (M) |
|---|---|---|---|---|---|
| | val | testA | testB | | |
| Full-Tuning | 65.1 | 68.1 | 61.4 | 64.9 | 149.97M |
| Fix Backbone | 74.9 | 77.1 | 72.0 | 74.7 | 0.00 M |
| Adapter (Houlsby et al., 2019) | 71.2 | 73.3 | 68.3 | 70.9 | 1.98M |
| Compacter (Karimi Mahabadi et al., 2021) | 73.9 | 75.8 | 70.8 | 73.5 | 1.62M |
| LoRA (Hu et al., 2021) | 73.4 | 75.7 | 70.2 | 73.1 | 1.57M |
| ETRIS (Xu et al., 2023) | 74.5 | 76.5 | 72.9 | 74.6 | 1.38M |
| DCRIS-B (Ours) | 75.8 | 77.7 | 72.9 | 75.5 | 1.34M |
| DCRIS-B (Ours) (Default Setting) | 76.4 | 77.9 | 73.7 | 76.0 | 2.69M |

and DCRIS-L both outperform existing parameter-efficient training methods such as ETRIS and BarLeRIa. Specifically, DCRIS-B achieves a substantial improvement, and DCRIS-L further enhances performance, demonstrating the effectiveness of our method. As shown in Table 2, our model maintains high efficiency with a relatively small number of training parameters, providing a compelling balance between performance and computational efficiency.

*Table 3.* Ablation study on the components of DCRIS. DA stands for DenseCross Adapter, and TA denotes Text Adapter.

| DA | TA | RefCOCO | | |
|---|---|---|---|---|
| | | val | testA | testB |
| ✗ | ✗ | 74.9 | 77.1 | 72.0 |
| ✓ | ✗ | 75.4 | 77.4 | 72.2 |
| ✗ | ✓ | 75.9 | 77.5 | 72.7 |
| ✓ | ✓ | 76.4 | 77.9 | 73.7 |

*Table 4.* Ablation study of different dimensions (Dim) of cross-dense adapters and text adapter.

| Visual Dim | Text Dim | Params | RefCOCO | | |
|---|---|---|---|---|---|
| | | | Val | TestA | TestB |
| 64 | 64 | 1.91M | 76.21 | 78.04 | 73.78 |
| 128 | 64 | 2.69M | 76.42 | 77.91 | 73.65 |
| 128 | 128 | 3.09M | 76.01 | 78.28 | 73.52 |
| 256 | 128 | 4.83M | 76.40 | 78.04 | 73.42 |

We also conducted comparisons on mixed RefCOCO datasets to verify the generalization ability of our models. As shown in Table 1, our DCRIS-L model achieves the highest average IoU score of 76.1, surpassing other state-of-the-art methods such as PolyFormer-L and UNINEXT-L. This demonstrates the robustness and effectiveness of our method as the data volume increases. Our proposed DCRIS models achieve significant improvements over existing RIS methods, both in terms of IoU and parameter efficiency.

## 4.4. Ablation Study

**Comparison with Other Parameter-Efficient Training Methods.** We conduct experiments comparing our Dense-Cross Adapter and Text Adapter (DCRIS-B) approach with other parameter-efficient training methods using DINO-B as the backbone. To ensure fairness, we retain the original parameter settings from previous methods and adjust the rank size to achieve comparable parameter counts. As shown in Table 2, our DCRIS-B method shows superior performance while maintaining efficiency. Adapter achieves an average accuracy of 70.9% with 1.98 million parameters. Compacter achieves 73.5% with 1.62 million parameters. LoRA achieving 73.1% with 1.57 million parameters. ETRIS reaches 74.6% with 1.38 million parameters. By adjusting the number of DenseCross Adapters from the default setting of 6 to 3, our DCRIS-B model surpasses these methods by achieving 75.5% with 1.34 million parameters, which is 0.9% of backbone parameters, even with a smaller parameter count than ETRIS. When the parameter count is further increased to 2.69 million (default setting), DCRIS-B achieves the highest average score of 76.0%, which is only 1.8% of backbone parameters. These results suggest that previous parameter-efficient training methods may heavily rely on the vision-text alignment capability of pre-trained models, which may not be as effective for the RIS downstream task. In contrast, our method does not have this limitation and can effectively enhance model performance. This demonstrates the robustness and efficiency of our approach in enhancing performance with a modest increase in parameters.

**Effect of Adapter's layers and dimensions.** To determine the optimal configuration of the DenseCross Adapter and Text Adapter, we conducted an ablation study varying both the number of layers and the dimensions of the adapters. Firstly, we evaluated the impact of different numbers of adapter layers. In this experiment, the visual DenseCross Adapter and the textual Text Adapter were inserted at the same layers. The results, shown in Table 5, indicated that

*Table 5.* Ablation study of different configurations of crossdense adapters and text adapters. For the "Position", we list the i-th layers that insert adapters in the backbone. For the "Percentage", we calculate the percentage of updatable parameters to the fixed backbone.

| Layers | Position | Params | Percentage | RefCOCO Val | TestA | TestB |
|---|---|---|---|---|---|---|
| 1 | [8] | 0.45M | 0.30% | 75.40 | 77.19 | 72.04 |
| 2 | [6,11] | 0.90M | 0.60% | 75.47 | 77.41 | 72.54 |
| 3 | [4,8,11] | 1.35M | 0.90% | 75.78 | 77.65 | 72.86 |
| 4 | [3,6,9,11] | 1.80M | 1.20% | 75.98 | 77.85 | 73.34 |
| 5 | [3,5,7,9,11] | 2.25M | 1.50% | 76.22 | 77.88 | 73.48 |
| 6 (default setting) | [1,3,5,7,9,11] | 2.69M | 1.79% | 76.42 | 77.91 | 73.65 |
| 7 | [1,3,5,7,9,10,11] | 3.14M | 2.09% | 76.19 | 78.07 | 73.41 |
| 8 | [1,3,4,5,7,9,10,11] | 3.59M | 2.39% | 75.71 | 77.87 | 73.34 |
| 12 | [0,1,2,3,4,5,6,7,8,9,10,11] | 5.39M | 3.59% | 76.11 | 78.07 | 73.49 |

performance generally improved as the number of layers increased from 1 to 6, with the best results achieved at 6 layers. Beyond 6 layers, performance gains plateaued and even slightly decreased, suggesting that additional layers did not contribute significantly to further improvement and, in some cases, led to overfitting.

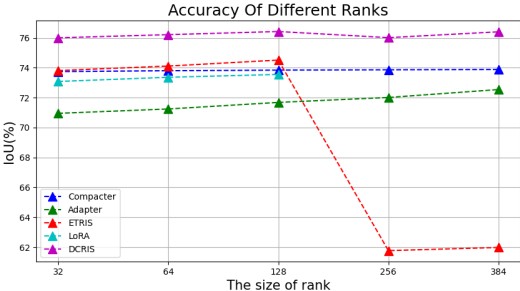

*Figure 2.* Ablation study of the Adapter's rank and comparison with other Parameter-efficient training Methods.

**Effect of DenseCross Adapter and Text Adapter.** We assessed the impact of the DenseCross Adapter (DA) and Text Adapter (TA) by performing an ablation study and reporting the results on validation and test datasets. From Table 3, it is evident that the baseline model without either adapter achieves the lowest average performance, with an average score of 74.7%. Introducing only the DenseCross Adapter yields a slight improvement, increasing the average performance to 75.0%. This indicates that the DenseCross Adapter enhances visual feature extraction. Applying only the Text Adapter results in a more significant improvement, with the average performance rising to 75.6%, demonstrating the Text Adapter's critical role in processing and integrating textual information. The combined use of both adapters produces the highest performance across all datasets, with an average score of 76.0%. This synergistic effect under-

scores the importance of integrating both visual and textual adaptations to enhance the model's capability in cross-modal feature learning and segmentation performance. The notable improvement, particularly on the testB dataset, highlights the robustness and generalization ability provided by the combined approach.

**Effect of varying the dimensions of the adapters.** We investigated the effect of varying the dimensions of the adapters. The results of this study are illustrated in Figure 2. In this experiment, the adapters were fixed at the optimal layers identified in the previous experiment [1,3,5,7,9,11] while varying their dimensions. Configurations tested included combinations such as 64,64; 128,64; 128,128; and 256,128. The results, presented in Table 4, showed that the combination of 128,64, which corresponds to the chosen parameter setup for DCRIS-B, proved to be the most effective. Although increasing the dimensions to 128,128 or 256,128 introduced more parameters, the performance improvements were marginal and did not justify the additional computational cost.

## 5. Conclusion

In this work, we propose a novel parameter-efficient training (PET) method for multi-modal dense prediction tasks, especially in referring image segmentation. Specifically, we adapt the pre-trained DINO model for referring image segmentation (RIS) by introducing a simple yet effective fine-tuning strategy. Our key innovation is the Dense-CrossAdapter, which can be seamlessly integrated into the visual backbone to improve fine-grained visual-text alignment. We also propose using text adapters to enhance the language encoder's capabilities. Our streamlined approach not only surpasses the performance of fully fine-tuned models but also does so more efficiently in terms of scalability and parameter management.

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

# A. Appendix

## A.1. Qualitative Analysis

As depicted in Figure 3, our visualization results compared to ETRIS highlight the superior accuracy of our DCRIS-B model in segmenting target objects. This enhancement in dense prediction and alignment between images and text is credited to the incorporation of the DenseCross Adapter and Text Adapter.

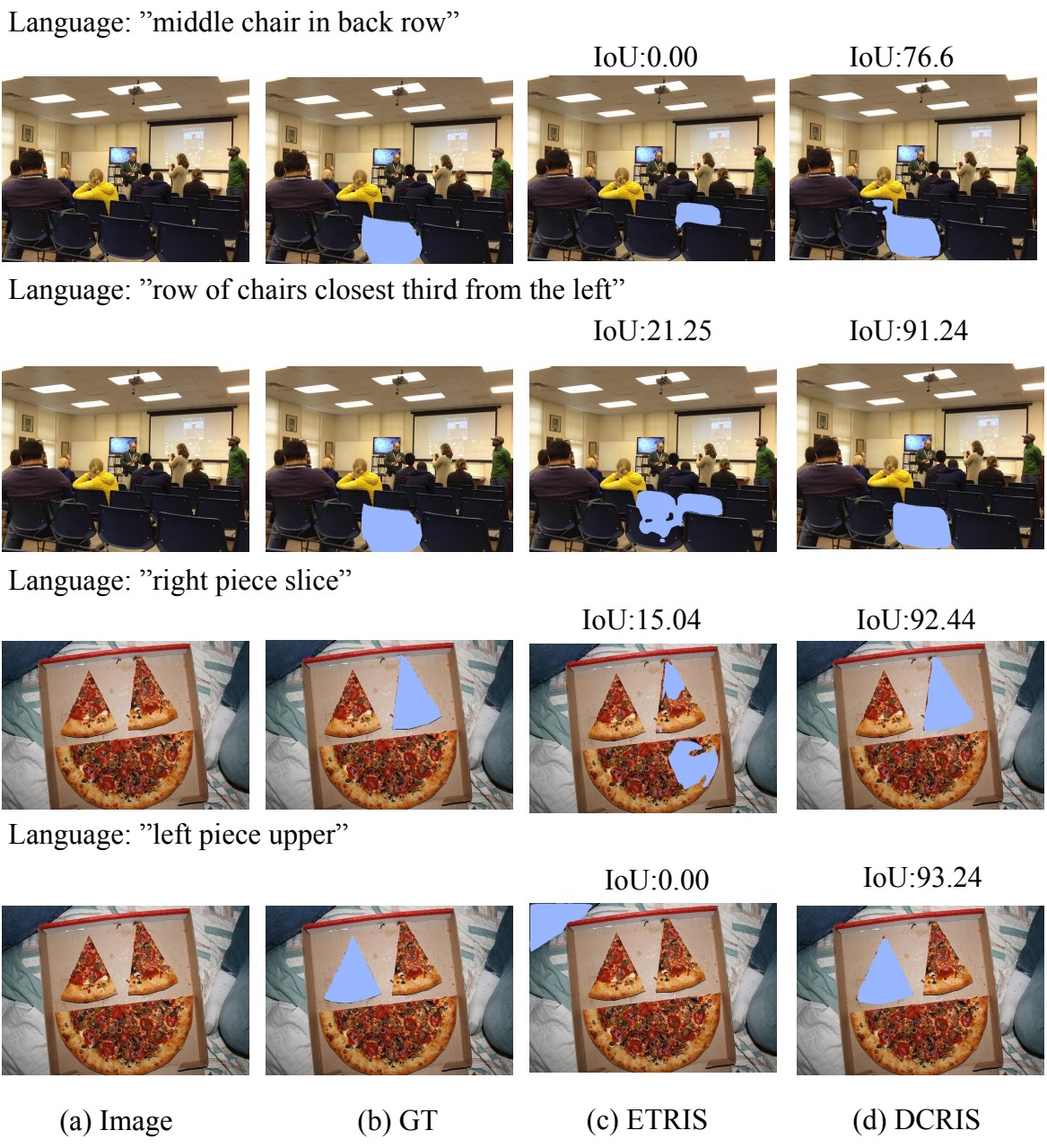

*Figure 3.* Comparison between DCRIS and state-of-the-art PET RIS method ETRIS.

As shown in Figure 4, we present qualitative results with different settings across various scenarios. In the figure, (c) represents the baseline ETRIS method; (d) shows our proposed DCRIS-B that only utilizes the Text Adapters; (e) illustrates our proposed DCRIS-B that only utilizes the DenseCross Adapter; (f) demonstrates our proposed DCRIS-B using both adapters; and (g) displays DCRIS-L trained on mixed datasets. In the first two rows of Figure 4, representing the easy

scenario, all methods can segment objects correctly. However, the details and finesse of the contours vary. Our proposed DCRIS-B and DCRIS-L achieve the best segmentation IoU, while ETRIS performs the worst among the compared settings. For the challenging scenario, depicted in the third and fourth rows of Figure 4, ETRIS is unable to correctly localize the object. In contrast, our proposed DCRIS-B and DCRIS-L accurately segment the target objects, demonstrating better dense prediction and improved image-text alignment capabilities due to the DenseCross Adapter and Text Adapter.

Language: "open book"

Language: "farthest left elephant"

Language: "man with glass in his hand"

Language: "pinkered donut"

Language: "the computer screen"

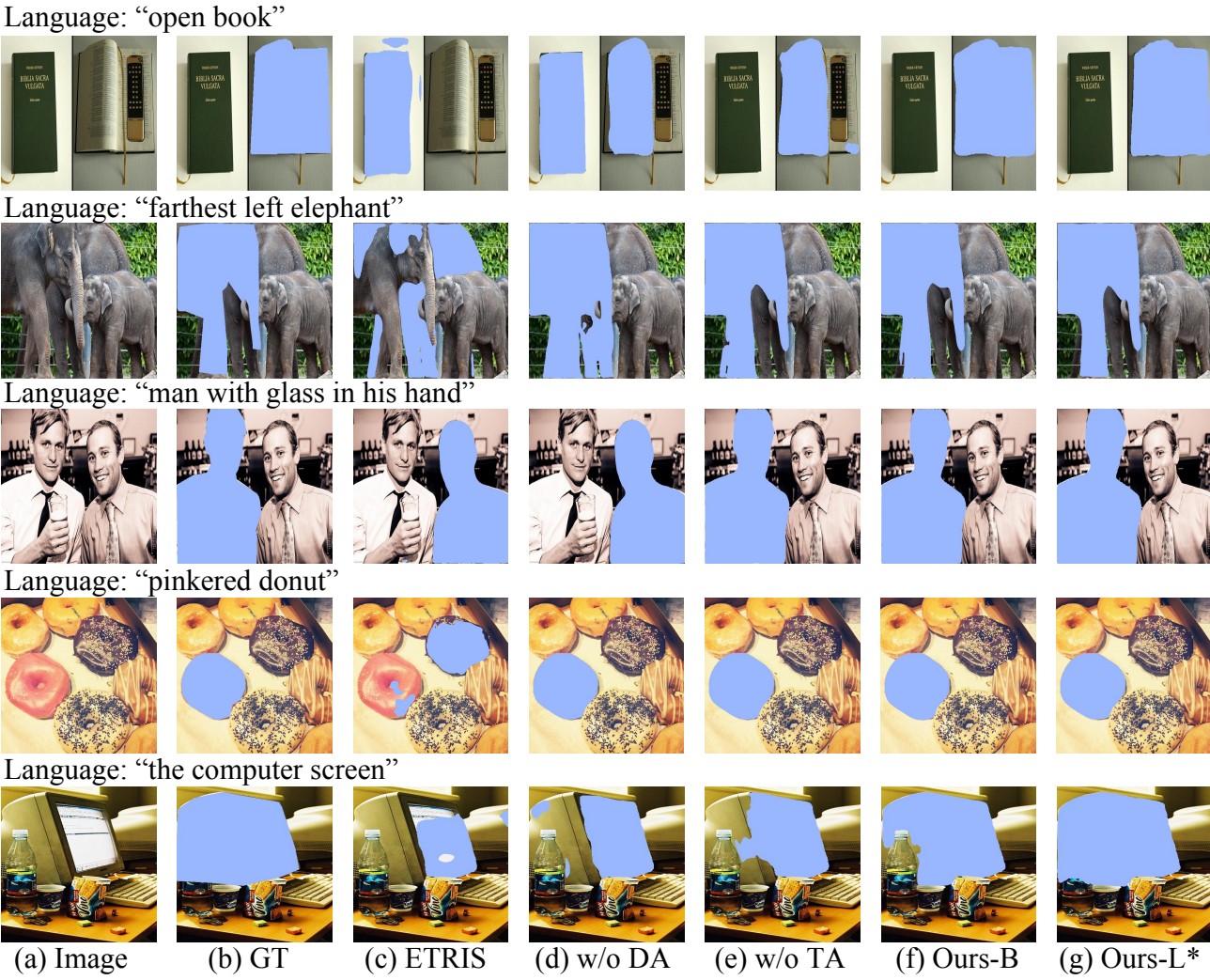

|  (a) Image  |  (b) GT  |  (c) ETRIS  |  (d) w/o DA  |  (e) w/o TA  |  (f) Ours-B  |  (g) Ours-L* |

Figure 4. Qualitative results: (a) the input image; (b) the ground truth; (c) ETRIS; (d) DCRIS-B using Text Adapter without DenseCross Adapter; (e) DCRIS-B using DenseCross Adapter without Text Adapter; (f) our proposed DCRIS-B; (g) DCRIS-L using mixed datasets. Best viewed in color.

## A.2. Limitations

While our proposed DCRIS has outperformed fully fine-tuned models in terms of efficiency, scalability, and parameter management, our experiments have been limited to the RIS task. Future work should extend the validation to other multi-modal segmentation tasks to further confirm the versatility of our approach. Moreover, as multi-modal large-scale models advance, exploring open-vocabulary zero-shot referring image segmentation presents a promising avenue for research.

