# OpenReview forum: "Enhancing Fine-grained Multi-modal Alignment via Adapters: A Parameter-Efficient Training Framework for Referring Image Segmentation"
_ICML.cc/2024/Workshop/WANT — WANT@ICML 2024 Poster_

### Official Review · Reviewer_SuW5 · 2024-06-11

**Confidence:** 3

**Summary:**

This study investigates the efficient training problem on referring image segmentation. It proposes DenseCrossAdapter, a parameter-efficient module designed to enhance low-rank visual feature propagation, and suggests using text adapters to improve textual features.

Pros:

- It uses DINO as the vision backbone and demonstrates the visual-text alignment capability of the proposed method.
- It tests the effectiveness of the proposed method on three challenging benchmarks.
- Systematically compare parameter-efficient training to full fine-tuning.
- Comprehensive ablation study.

Cons:

- It will increase the inference time since the adapters cannot be merged into the model parameters. Add some inference efficiency metrics such as inference time/latency.

**Strengths:**

- Systematically compare parameter-efficient training to full fine-tuning.
- Comprehensive ablation study.

**Weaknesses:**

- lack analysis of inference efficiency

---

### Official Review · Reviewer_Ki4k · 2024-06-13
**Using adapters to further enhance multimodal alignment for image segmentation**

**Confidence:** 2

**Summary:**

This paper proposes to use lightweight adapters for both vision and text tasks to further improvement the multimodal alignment for the referring image segmentation tasks. The design is simple that DenseCrossAdaptors and TextAdapters are added to the vision and text towers, respectively, trained together with contrastive loss. The proposed method shows superior results over the previous literature.

**Strengths:**

1. The idea of adding adapters to vision/text towers are very straightforward and easy to understand.
2. The final results are very impressive.

**Weaknesses:**

It seems the design of the two adapters are different. It seems the choice of them (like why not using DenseCross adapter for text) is not discussed in the ablation.

---

### Meta-Review · Area_Chair_Wvmg · 2024-06-17

**Recommendation:** Accept (Poster)
**Confidence:** 3

**Metareview:**

The manuscript introduces a simple yet effective Parameter-Efficient Training method for multi-modal models, namely the multi-modal dense prediction tasks. All reviewers appreciate the impressive results.

---

### Decision · Program_Chairs · 2024-06-17

**Decision:**

Accept (Poster)

**Comment:**

We thank the authors for their time and contribution to WANT and we are pleased to share that after the reviewing process the paper has been accepted. Congratulations! We encourage the authors to consider reviewers' feedback for the improvement of the camera-ready version. We hope to see you in person at the workshop and brainstorm on efficient training research together!